# Enzymatic Extraction and Characterization of Pectin from Cocoa Pod Husks (*Theobroma cacao* L.) Using Celluclast^®^ 1.5 L

**DOI:** 10.3390/molecules26051473

**Published:** 2021-03-09

**Authors:** Licelander Hennessey-Ramos, Walter Murillo-Arango, Juliana Vasco-Correa, Isabel Cristina Paz Astudillo

**Affiliations:** 1GIPRONUT, Departamento de Química, Facultad de Ciencias, Universidad del Tolima, Ibagué 730006, Colombia; 2Área de Agroindustria, Servicio Nacional de Aprendizaje—SENA, km 5, vía El Espinal—Ibagué, Dindalito 733527, Colombia; 3Department of Agricultural and Biological Engineering, Penn State University, State College, PA 16802, USA; jpv5237@psu.edu; 4Facultad de Ingeniería Agronómica, Universidad del Tolima, Ibagué 730006, Colombia; icpaza@ut.edu.co

**Keywords:** cellulase, optimization, byproducts, agricultural waste

## Abstract

Cocoa pod husks are a waste generated during the processing of cocoa beans. We aimed to explore the enzymatic extraction of pectin using cellulases. The extraction process was optimized using a central composite design (CCD) and analyzed by response surface methodology (RSM). The parameters optimized were feedstock concentration (%), enzyme dosage (µL/g), and time (h). Three dependent variables were studied: pectin yield (g/100 g dry husk) (R^2^ = 97.02), galacturonic acid content (g/100 g pectin) (R^2^ = 96.90), and galacturonic acid yield (g/100 g feedstock) (R^2^ = 95.35). The optimal parameters were 6.0% feedstock concentration, 40 µL g^−1^ of enzyme, and 18.54 h, conditions that produced experimentally a pectin yield of 10.20 g/100 g feedstock, 52.06 g galacturonic acid/100 g pectin, and a yield 5.31 g galacturonic acid/100 g feedstock. Using the chemical extraction method, a yield of 8.08 g pectin/100 g feedstock and a galacturonic acid content of 60.97 g/100 g pectin were obtained. Using assisted sonication, a pectin yield of 8.28 g/100 g feedstock and a galacturonic acid content of 42.77 g/100 g pectin were obtained. Enzymatically optimized pectin has rheological and physicochemical features typical of this biomaterial, which provides an interesting alternative for the valorization of cocoa husks.

## 1. Introduction

Cacao (*Theobroma cacao* L.) is a crop of high commercial importance that generates an income for more than 4.5 million families all over the world [1]. In 2019, cacao grain production worldwide was approximately 4.82 million tons. Ivory Coast, Ghana, and Ecuador are the three main producers in the world with a 70% global market share [2]. Since the commercial product of interest from the cacao fruit is the dry grain, it is estimated that in order to produce 1 ton of dry grain, about 10 tons of cacao husk are generated. Unfortunately, this significant amount of residue is not utilized, despite being a potential feedstock for the extraction of added value compounds [3,4], thus being left out on the ground to decompose causing bad smells, soil chemical imbalance, and becoming a potential source for the development of pathogenic microorganisms that affect crop health [5].

Due to the chemical composition of cacao husk, which includes fiber, phenols, carbohydrates, lignin, protein, and minerals, it has been used as feedstock in conversion processes to obtain pectins [6,7], make briquettes [8], extract phenolic [4], produce biofertilizers and activated carbon, and as fiber source for animal feeding [9]. Pectins are a diverse group of polysaccharithe fluid is considered pseudo plasticdes of d-galacturonic acid with units linked by α 1–4 bonds that also contain other sugars such as galactose, arabinose, glucose, and xylose. Pectin extraction has generated increased interest in the cacao agroindustry [10,11]. Pectin is widely used in food and pharmaceutical industries due to its particular gelling properties and health benefits, including regulation of appetite, coronary disease prevention, and certain types of cancer mitigation [12,13].

Among the pectin extraction methods, one of the most highlighted is chemical extraction with citric, nitric, and hydrochloric acid. The main pectin properties studied are those that allow the determination of its potential uses such as yield, galacturonic acid content, and rheological properties [14,15,16]. Different extraction conditions and the influence of different process parameters on the pectin properties have been explored, aiming to optimize the pectin extraction conditions from different agroindustrial waste through response surface designs [17,18,19]. Emerging extraction methods have also been explored such as extraction with subcritical water [20]. However, there have been few studies oriented to extraction using biotechnological methods, such as the use of enzymatic complexes [21]. Enzymatic methods have a high industrial potential for pectin extraction, since pectins with similar characteristics can be obtained and with even better yields as compared to the traditional chemical methods. In addition, the extraction conditions are significantly milder, and the residues generated in the extraction process are less polluting. Besides, the enzymatically de-pectinized material can be used for the production of other added-value products [22,23].

Considering that there is a market tendency that demands products labeled as completely natural or biobased, there are combined efforts from the industry to achieve their production [24,25]. This is how the concept of biorefinery became relevant to take advantage of the fact that food and agricultural waste are reservoirs of carbohydrates, proteins, lipids, and metabolites with nutraceutical characteristics and that it is possible to turn them into commercial products, thus generating added value [26]. Additionally, natural products are of great interest for the food and pharmaceutical industries, which is reflected on the pectin world market, which was about USD 1.0 billion in 2019 and it is estimated to reach USD 1.9 billion in 2025 [27].

Cellulases, which is a group of enzymes with hydrolytic properties that degrade cellulose, have been use for pectin extraction from different types of agroindustrial residues such as artichoke, apple pomace, sisal, and lime peels, producing pectins with higher contents of galacturonic acid at yields comparable to those of the conventional extraction, which uses inorganic acids and high temperatures [28,29,30,31]. However, it is necessary to study whether the use of cellulases in substrates such as cacao pod husks can induce cellulose hydrolysis and subsequently generate pectin release, and to determine how quality parameters such as galacturonic acid content and yield are affected.

The main purpose of this study is to use the main residue generated in the cacao fruit processing, which is the cacao pod husks, to obtain an added-value biocompound such as pectin. Pectin extraction from cacao pod husk powder (CPHP) using the commercial enzyme Celluclast^®^ 1.5 L is optimized using a central composite design (CCD) with three independent variables: cacao pod husk concentration, enzyme dose, and time. Three dependent variables are studied: pectin extraction yield (EY), galacturonic acid (GA) content, and yield of galacturonic acid (YGA). The design is analyzed by response surface methodology (RSM). Results of the optimized enzymatic extraction are compared with those from extraction with citric acid and assisted by ultrasound. A chemical and rheologic characterization of the pectin extracted from CPHP is performed to evaluate the influence of the enzymatic method on the pectin properties. In addition, a CIELab scale color characterization of the enzymatically extracted pectin versus a commercial pectin is performed.

## 2. Results and Discussion

### 2.1. Chemical and Ultrasound-Assisted Pectin Extraction

Pectin EY obtained by the chemical method with citric acid was 8.08 ± 0.8 g/100 g CPHP, and had a GA content of 60.97 ± 0.51 g/100 g pectin. Under similar conditions, yields around 9.0 to 12.2 g/100 g CPHP, and GA contents around 31.19 to 65.08 g/100 g pectin have been reported [14,16,32]. According to Vriesman et al. (2012), the maximum GA in cacao pectins extracted with citric acid under similar conditions (pH 3.0, 95 min, 95 °C) was 65.1 GA g/100 g pectin [16], which is similar to the values obtained in the present study. Even though the extraction reported by Vriesman et al. (2012) was performed under very similar conditions, the slight difference in GA content can be mainly attributed to the type of husk used [32]. It is also necessary to consider that pectin EY and GA content of CPHP pectin are affected by pH, the type of acid used in the process, time, and temperature [14,33].

Through assisted sonication a pectin EY of 8.28 ± 0.42 g/100 g CPHP was obtained with a GA content of 42.77 ± 0.01 g/100 g pectin. Yield increased slightly with this method, managing to reduce time and temperature as compared to the acid extraction, generating similar results. Nevertheless, the GA content of this pectin was lower than that of the acid extracted pectin, which may be due to the lower time and temperature of the process. It has been shown that in assisted sonication treatments, pectins with higher GA content are obtained at higher temperature [34]. However, the main advantage of emerging methods such as sonication is that they can be operated at lower temperatures than traditional methods to reduce the energy cost of the process. Therefore, operating above 60 °C in order to increase GA content would not be feasible if there is an expectation of implementing this technology at an industrial scale. Other studies have chosen to use to sonication as a feedstock pretreatment to then combine them with enzymatic and microwave extraction methods, thus obtaining better results [30,35].

### 2.2. Enzymatic Pectin Extraction

The optimization of the experimental conditions for enzymatic pectin extraction was performed through RSM. Figure 1, Figure 2 and Figure 3 represent the contour graphs that relate the response values adjusted based on model equations (Table A2). Table A2 shows that the independent variables that were studied, EY, GA content, and YGA adjust to a regression model with R^2^ 97.02, 96.90, and 95.35, respectively. In addition, all models were significant (*p* < 0.05). Ra^2^ values and R^2^ are very close, indicating an adequate correlation between predicted and experimental values [36]. There were significant linear and quadratic effects in all models, as well as interactions of the variables studied.

According to the CCD applied, pectin EY was between 6.46 and 11.31 g/100 g CPHP with a higher EY at a solid loading of 5.18% of CPHP, enzyme loading of 58.04 µL/g, and 20.35 h of extraction time (Table A1). The yield increased with the concentration of CPHP and time. On the other hand, the best EYs were achieved at low enzyme concentrations (Figure 2, Table A1). Similar studies with other feedstocks such as apple pomace found that using doses of 50 µL/g of Celluclast^®^ 1.5 L produced pectin yields of about 19% [37]. Similarly, it has been found that doses of 55 µL/g Celluclast^®^ 1.5 L on sisal waste reached pectin yields of 9.4% [30]. As for the time, it is possible to say that Celluclast^®^ 1.5 L requires long periods of time to hydrolyze the cellulose of the cell wall effectively and extract the pectin [38], which is why the best EYs according to the optimization are obtained at around 20.35 h. With these results, it is confirmed that cellulose hydrolysis of CPHP allowed the release of pectin molecules.

GA content of these pectins was between 39.43 and 54.39 GA g/100 g pectin, and the higher GA content was obtained at 2.81% of CPHP, 110.96 μL/g of enzyme, and 20.35 h (Table A1). Figure 2 indicates that purity of pectin expressed in GA content increased favorably both with low concentrations of enzyme and CPHP. On the contrary, GA content increased as time increased. The FAO establishes that pectins must be rich in galacturonic acid and have a value over 65 GA g/100 g pectin [39]. Pectins below this value are associated with the possible presence of proteins, starches, ash, and other sugars that precipitate next to the pectic gel [40]. Such 65 GA (g/100 g pectin) value is set to ensure gelling capacity of pectin, since pectic gel forms when the region of the homogalacturonan chains, which are formed mainly by galacturonic acid units, intertwine over each other, creating a three-dimensional crystalline network where solutes such as sucrose and water are trapped [41].

The GA content of the pectin extracted by the enzymatic method was lower than that of the pectin obtained by the chemical method. It has been shown that extracting pectin at lower pH increases GA content favorably [42]. This matches the results obtained given that the chemical method was performed at a pH 2.5 and the enzymatic method at pH 4.6. On the other hand, enzymatic methods can induce atypical compound solubilization during pectin extraction, which can be reflected in low GA content. Nevertheless, despite having a lower GA content, enzymatically extracted pectins have shown a rheological behavior similar to that of pectins extracted by acid methods [43], which is why it is necessary to evaluate gelling capacity of the obtained CPHP pectin to determine its potential use.

Other studies have found GA content of CPHP pectins between 31.19 to 65.20 g/100 g pectin, depending on extraction conditions [14]. It is desirable to obtain a pectin with high GA content, which is strongly influenced, as the results showed, by the extraction conditions. However, the pectin GA content also depends to a great extent on the purification process for which ethanol has been traditionally used, even though ethanol precipitates other components that are not desirable in pectin [44]. However, ethanol is widely used because it adjusts favorably when scaled industrially [45].

YGA found was between 2.80 and 5.23 g/100 g CPHP, and the highest YGA value was obtained at 4% of CPH, 40 µL/g of enzyme dose, and 15 h (Table A1). YGA improved with the increase of extraction time and CPHP concentration. On the contrary, reducing enzyme dosage generated an increase in YGA (Figure 3). Since YGA considers two variables, namely pectin EY and pectin quality measured in terms of GA content [18], it is considered a good measure to be optimized. When performing response optimization, the function of individual desirability of YGA provides the following optimal point: 6.0% of CPHP, 40 µL/g of enzyme dose, and 18.54 h for a YGA of 6.01%. When validating these points experimentally, an EY of 10.20 g/100 g CPHP, a GA content of 52.06 g/100 g, and YGA of 5.31 g/100 g CPHP were obtained.

Thus, it is shown that the enzymatic method with Celluclast^®^ 1.5 L is an appropriate alternative for pectin extraction from CPHP with even better yields than acid extraction and assisted sonication. Therefore, it can be used to replace the chemical hydrolysis extraction method, which operates at high temperatures and very acid pH, thus becoming an environmentally friendlier alternative. Although pectin yields of the enzymatic method are similar to those of traditional methods, GA content results are lower for the enzymatic method, which is why it becomes necessary to make later improvements to increase these values.

### 2.3. FTIR Spectra, Degree of Esterfication and Total Sugars of Enzymatically Extracted Pectin

The presence of bands between 3300 and 3500 cm^−1^ that can be identified in the FTIR spectra (Table 1, Figure A1) indicates that hydroxyl groups (O-H) are present in the pectin structure [46]. Signals around 3000 and 2800 cm^−1^ correspond to the tension of C-H due to vibration of the methyl ester group (CH_3_). The bands around 1750 and 1650 cm^−1^ correspond to the esterified carboxyl group (COO-) and non-esterified carboxyl groups (COO-R) of pectin, respectively [47]. Bands between 1623 and 1428 cm^−1^ correspond to wavelength features for polygalacturonic acid [46]. This indicates that the pectin obtained under the optimized enzymatic extraction conditions was rich in polygalacturonic acid. Signals between 1200 and 950 cm^−1^ are characteristic for polyssacharides. Therefore, it is considered as the chemical print of these compounds [48].

The degree of esterification of the pectin extracted in the optimal conditions was 24%. Pectins with a degree of esterification below 50% are considered low-methoxyl pectins. These pectins usually form gels under the presence of calcium ions and a wide range of pH [49]. However, Vriesman et al. (2012) obtained a pectin from cocoa pod husks with a 40.3% degree of esterification and when formulating the gel in the presence of calcium ions and low pH, no gelification was observed [16].

The total sugars content of the pectin enzymatically extracted under the optimized conditions was 87.49 g/100 g pectin, which is a direct indication of its purity. Vriesmann (2011) reported a total sugar content of 69.9 g/100 g in CPHP pectin extracted by the acid method [15]. The high sugar content obtained is an indication the efficiency and selectivity of the pectin extraction by the enzymatic method [50].

### 2.4. Rheological Characterization and Gelling Test

Pectin rheograms were adjusted to the Ostwald–de Waele model (Equation (1)),
(1)$$\eta =k\gamma^{n-1}$$
where
η is apparent viscosity, *k* is the consistency 
coefficient (Pa·s *^n^*), γ (s^−1^) is the shear stress, 
and n is the flow index [51]; and *k* and n values describe fluid behavior [52]. Consistency index *k* is proportional to the 
sample viscosity and the n value describes flow behavior, indicating that 
if n < 1, the fluid is considered pseudo plastic; 
if n > 1 the fluid is dilating; and if n = 1 is a Newtonian fluid [52,53]. Values of n obtained (Table 2) indicated that pectins tested at different concentrations showed a non-Newtonian 
flow and a pseudo-plastic behavior since the value of n is smaller than 1. It is observed that pectin viscosity decreased when shear stress increased (Figure 4). The consistency index showed that if samples with high viscosity are desired, it is necessary to increase pectin concentration.

Gelling test (Figure A2) showed that pectin extracted by the optimized enzymatic method can form a gel with viscoelastic properties, considering that CPHP pectin gel managed to be contained in the interior of the beaker the same way as the gel of commercial pectin. The gelling conditions used for this test are typical of high-methoxyl pectins and included low pH and high sucrose concentration, even though the obtained CPHP pectin was low methoxyl. Gelling of low-metholxyl pectins usually requires calcium ions. This is an interest characteristic of the obtained CPHP pectin, which behaves like high-metholxyl pectins despite its low degree of esterification. This behavior has been reported before in pectins obtained from cacao residues [6]. Thus, CPHP pectin could be used in products with high acidity and high solids content such as jams.

### 2.5. Colorimetry of Pectin and Phenolic Content of CPHP

Pectin color plays an important role in the final appearance of the product to which it is added, and pectin extraction should not affect the color of the final product in a negative way. Therefore, the pectin purification process is fundamental to remove any pigments that interfere with this quality parameter. The color of the commercial pectin in CIELab scale was L = 77.13, a = 2.17, and b = 14.46 and the color of the CPHP enzymatically extracted pectin under optimized conditions was L = 69.71, a = 4.33 and b = 17.48. Delta-E value was 8.3. Values over 5 indicate that there are noticeable variation to the human eye [54,55]. Therefore, CPHP pectin showed color variations at plain sight when contrasted with commercial pectin. The luminosity parameter L indicates that cacao pectin is slightly darker than commercial pectin and values of a and b were a somewhat higher when compared to those of the commercial pectin. This variation is due to difference in feedstock, since the exocarp of cacao pod husks has a high pigment content related to mostly to phenolic compounds, which could transfer to the pectin.

Measurement of CPHP phenolic content was 988.09 ± 75.47 (mg GAE/100 g), and after the pectin extraction, the phenolic content of the leftover material was 413.3 ± 31.10 (mg GAE/100 g). This indicated that an important amount of phenolic compounds was transferred to the extracting solution, which presents an interesting challenge that can lead to future research to recover these added-value compounds. Similar studies have shown pectins with visible color occur due to the presence of polyphenols [56], and that during the pectin precipitation phase yellowing (+a) and reddening (+b) indexes increase, and luminosity (L) values decrease [57].

## 3. Materials and Methods

### 3.1. Feedstock

Cacao pod husks were provided by the Servicio Nacional de Aprendizaje—SENA (National Service of Learning, Espinal, Tolima, Colombia). These husks were washed with distilled water, cut in small pieces, and dried at 48 °C for 24 h. Dry husks were ground and sieved through a 500 µm sieve and stored at 6 °C.

### 3.2. Enzymatic Pectin Extraction

Enzymatic pectin extraction was optimized through a CCD with three independent variables: CPHP concentration (%), enzyme dosage (µL/g), and time (h). Three dependent variables were studied through RSM: pectin extraction yield (g pectin/100 g CPHP), galacturonic acid content (g GA/100 g pectin), and galacturonic acid yield (g GA/100 g CPHP). To determine the influence of the independent variables on the responses, the results were adjusted to a second order polynomial model. Model adjustment was tested with the R-squared and adjusted R-squared coefficient of determination. Model and coefficient significance were determined by an analysis of variance (ANOVA). Response optimization was made with the function of individual desirability for the galacturonic acid yield (YGA) variable. The software Minitab v.17 (Minitab Inc., State College, PA, USA) was used for these analyses.

Enzymatic pectin extraction from CPHP was performed with the commercial enzyme Celluclast^®^ 1.5 L (Novozymes, Bagsværd, Denmark). The process was performed in an orbital shaker at 50 °C, 200 rpm and 50 mM sodium citrate buffer pH 4.6. After hydrolysis, the material was filtered through cheesecloth, then centrifuged at 4000× *g* for 5 min at 4 °C. The supernatant was mixed with 4 times the volume of 96% ethanol and cooled at 4 °C for 24 h. The pectic material in gel form that was obtained after the cooling was washed with twice the volume of 70% ethanol at 4 °C for 2 h and centrifuged again. The material obtained was dried at 48 °C in an oven until reaching constant weight.

Extraction yield (EY) was defined as the pectin dry weight over the weight of CPHP (g pectin/100 g CPHP). Galacturonic acid content (GA) was used as an indirect measurement of the pectin purity and it was defined as the weight of GA obtained as described in Section 3.5 over the dry weight of pectin (g GA/100 g pectin). Galacturonic acid yield (YGA) relates the EY and the GA content, as shown in Equation (2) [18]:(2)$$\mathrm{YGA}=\frac{\mathrm{EY}\;\times \;\mathrm{GA}}{100}$$

### 3.3. Acid Pectin Extraction

Chemical extraction of pectin from CPHP was performed with a citric acid solution at pH 2.5 for 95 min at 95 °C with a solid loading of 0.04 g/mL in a reflux system with agitation at 200 rpm. These values were chosen because they correspond to the optimal conditions for the pectin extraction from cacao pod husks reported by other studies [16]. After hydrolysis, the material was filtered through cheesecloth and centrifuged at 4000× *g* for 5 min at 4 °C. Pectic material precipitation was carried out as in the enzymatic extraction, as described in Section 3.2.

### 3.4. Ultrasound-Assisted Pectin Extraction

Ultrasound-assisted pectin extraction from CPHP was performed with a 13 mm ultrasonic probe (VCX 750 Ultrasonic Microprocessor, Sonics & Materials, Newtown, CT, USA) operated at 20 KHz and a maximum output power of 750 W. The CPHP solid loading was of 0.04 g/mL in a citric acid solution at pH 2.5 and 60 °C, for 40 min at 90% amplitude with 10 s pulses. Pectic material precipitation was carried out as in the enzymatic extraction, as described in Section 3.2.

### 3.5. Pectin Characterization

Galacturonic acid was measured as the uronic acids content using the m-hydroxydiphenyl (mhdf) method, with a standard curve of GA monohydrate (0–0.1 g/L), as reported previously [58]. Degree of esterification was estimated by FTIR spectroscopy (Nicolet-400, Thermo Scientific, Waltham, MA, USA) with OMNIC 4.1 software, calculating the relation between the areas of the peaks 1750 cm^−1^ and 1610 cm^−1^, which correspond to the esterified carboxyl group and non-esterified carboxyl groups of pectin, respectively [47]. The discs were prepared in a proportion of 10:1 parts of KBr and pectin, respectively. The spectra were collected in a frequency rank of 4000–400 cm^−1^, with 100 scans and 8.0 cm^−1^ resolution. The total sugar content was determined by the phenol-sulfuric method [59].

### 3.6. Color Characterization

Color was measured in CIELab space coordinates, L (lightness: 0 = black, 100 = white), a* (−a = greenness, +a = redness) and b* (−b = blueness, +b = yellowness) in a Cr- 5 colorimeter (Konica Minolta, Tokyo, Japan). Equation (3) was used to determine the color variation of the optimized cacao pectin versus the commercial one [60]:(3)$$\Delta E=\;\sqrt{{\left(L_{0}-L_{i}\right)}^{2}+{\left(a_{0}-a_{i}\right)}^{2}+{\left(b_{0}-b_{i}\right)}^{2}}$$
where Δ*E*: color variation, $\;a_{0}-a_{i}$: variation from green to red between measurements,
$b_{0}-b_{i}$: variation from blue to yellow between measurements, 
and $L_{0}-L_{i}$: luminosity variation between the measurements.

### 3.7. Determination of Rheological Properties and Gelling Test

A DV-III Ultra rotational rheometer (Brookfield, Middleboro, MA, USA) was used to determine rheological properties of pectin solutions. Pectin solutions were prepared at different concentrations with distilled water (0.1, 1, and 2% *w*/*w*). The samples were analyzed at 25 °C at a shear rate between 9.3 and 130 s^−1^.

Gels were prepared using enzymatically optimized pectin and the commercial pectin (high-methoxyl pectin from citrus peel, Sigma-Aldrich, St. Louis, MO, USA) under regular conditions concerning °Bx and pH for jam elaboration according to current regulation in Colombia [61]. For this process, water sucrose solutions at 65 °Bx, 1% pectin, and 0.05% citric acid underwent heating with constant agitation until complete mixing was achieved. The mixture was let to rest at room temperature and then beakers were inverted to visually determine if the gel stays solid or flows down the beaker. This type of test has been used to visually determine the pectin gelling capacity in a qualitative way [62].

### 3.8. CPHP Phenolic Content Determination

Quantification of total phenols from cacao pod husks before and after the pectin extraction was performed by the Folin-Ciocalteu method [63]. The feedstock solid loading was of 0.1 g/mL and the solvent was a mixture (1:4, *v*:*v*) of distilled water and 96% ethanol. The samples were sonicated at 90% amplitude with a real power of 0.23 W/mL. Subsequently, the mixture was filtered and lyophilized. Total phenol content was quantified spectrophotometrically with a standard calibration curve of galic acid [64] in concentrations between 15.62 and 1000 µg/mL at 760 nm. Results were expressed as equivalent milligrams of galic acid per 100 g of extract (mg GAE/100 g).

## 4. Conclusions

In this study, it was possible to show that cacao pod husks can satisfactorily be used for pectin extraction with commercial enzymes such as Celluclast^®^ 1.5 L with similar or slightly better results when compared with acid hydrolysis and assisted sonication methods. The enzymatic method proposed allows the reduction of the extraction temperatures and the amount of acid use, which would implicitly reduce the environmental impact of the extraction process. Moreover, a de-pectinized leftover material is obtained with potential for use as feedstock to obtain other added-value compounds, such as phenolic compounds, which could be of importance for biorefinery development, process integration, and circular economies. The enzymatic extraction process, as compared with the traditional pectin extraction methods, generated higher pectin yields, which reached up to 10.20 g/100 g CPHP with a GA content of 52.06 g/100 g pectin. Pectin solutions exhibited non-Newtonian flow and a pseudo-plastic behavior typical of this type of biomaterials. The enzymatically extracted pectin could be classified as a low-methoxyl pectin according to its degree of esterification; however, it showed gelling properties like those of high-methoxyl pectins. These results are promising to use this pectin industrially. However, it is necessary to perform other studies to evaluate its potential use in food and other industrial products and to improve the pectin purity and color.

## Figures and Tables

**Figure 1 molecules-26-01473-f001:**
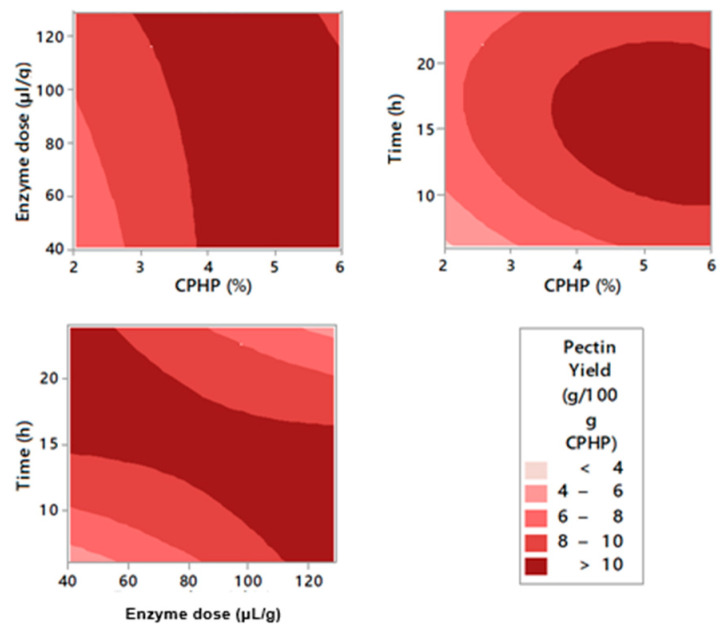
Contour plots for pectin extraction yield from cacao pod husks.

**Figure 2 molecules-26-01473-f002:**
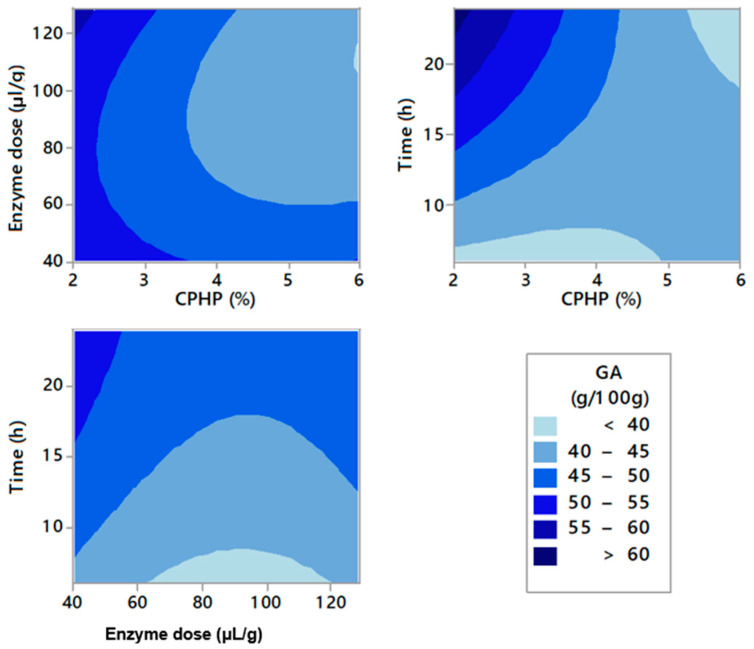
Contour plots for galacturonic acid contents of pectins extracted from cacao pod husks.

**Figure 3 molecules-26-01473-f003:**
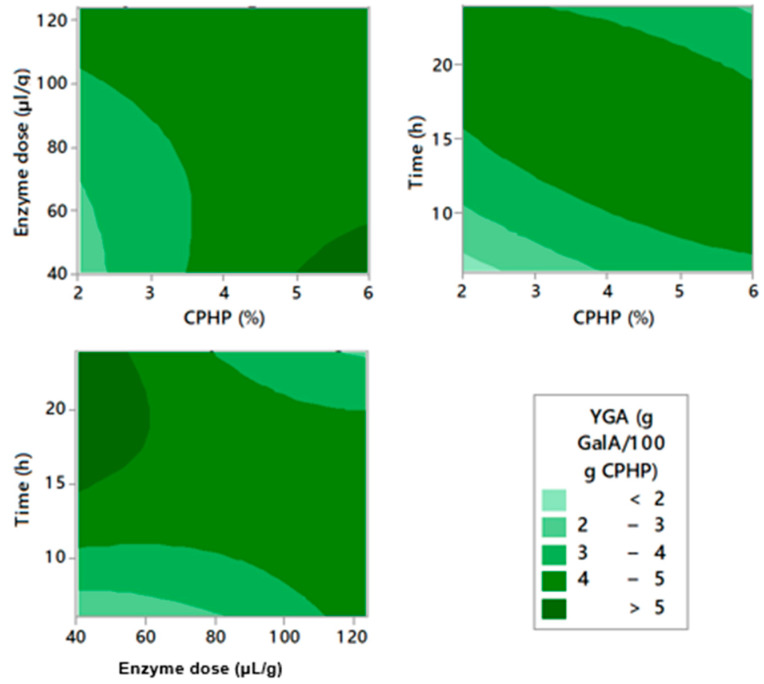
Contour plots for yields of galacturonic acid from cacao pod husks extraction.

**Figure 4 molecules-26-01473-f004:**
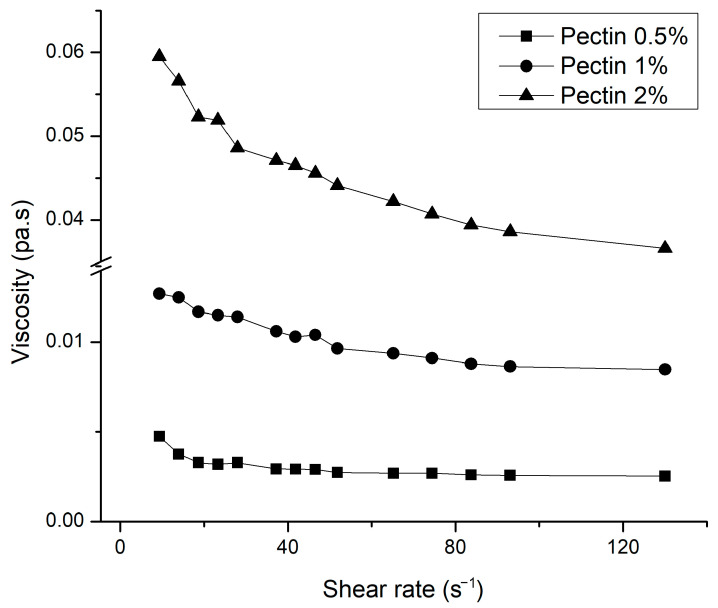
Flow behavior of extracted pectin of CPHP at different concentrations.

**Table 1 molecules-26-01473-t001:** FTIR wavenumber in the range 4000–3500 cm^−1^ of pectin obtained from cacao pod husk powder.

FTIR Wavenumber (cm^−1^)		Assignments
CPHP Pectin	Commercial Pectin	
3382	3437	O-H
2932	2953	C-H
1730	1748	C=O
1612–1421	1630–1445	C=O from free carboxyl groups
950–1200	Fingerprint	Fingerprint

**Table 2 molecules-26-01473-t002:** Pectin parameter determination adjusted to Ostwald–de Waele model.

Concentration (%)	K (Pa·sn)	*n*	R^2^
0.5	0.00549	0.83	0.95
1	0.02064	0.81	0.95
2	0.09957	0.78	0.98

## Data Availability

The data presented in this study are all available in the current article and Appendix A.

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
