# Peer review of "Enzymatic Extraction and Characterization of Pectin from Cocoa Pod Husks (Theobroma cacao L.) Using Celluclast® 1.5 L"

_molecules, 2021, doi:10.3390/molecules26051473_

Round 1
Reviewer 1 Report
The manuscript entitled “Enzymatic extraction and characterization of pectin from cocoa pod husks (Theobroma cacao L.) using Celluclast® 1.5L”, authored by Hennessey-Ramos and colleagues, deals with the optimization of the extraction procedure of pectins from Theobroma cacao using a commercial cellulase. Authors evaluated the effect of the changes of feedstock concentration (%), enzyme dosage (μl/g), and time (h), on pectin yield (g/100 g dry husk), galacturonic acid content, and galacturonic acid yield. The manuscript is very well written. The obtained data are clearly discussed and illustrated. Moreover, the discussion is supported by literature evidences, and the results are compared to previously published data.
I have only some little suggestions:
- “Cocoa pod husks were used as feedstock for pectin enzymatic extraction using Celluclast® 1.5L.” This sentence in the abstract is not useful. It is entirely contained in the title of the manuscript.
- keywords should be words not contained in the title of the manuscript. Their aim is to make the manuscript more visible when the universal research tools (such as PubMed) are employed. Some of the words provided by the authors are repetitive, as they appear both in the title and in the abstract. Please propose others and different keywords.
- Please check that Theobroma cacao is written in italics all over the manuscript, including the references
- Figure 3 appears to have smaller panels than the other figures.
- Are equation 1 and 2 reported as figures? Please, write them using equation tool of Microsoft word.
- I think that there is an error in equation 3: what does “a??E” means?
- Is it possible to colour the graphs of figure 1-3 differently basing on the type of parameter evaluated? it would certainly make the presentation clearer.
- In table 2, R2 should be R2
- In table 1, cm-1 should be cm-1
- Other typos may be present in the manuscript. So, I hardly encourage the authors to check it before submit the revised version of their manuscript.
Author Response
We greatly appreciate your suggestions, as they are of great importance to improve the manuscript and for the development of academic skills. The following table was made to answer each suggestion. Changes can be verified in the new version of the manuscript. Please see the attachment.

Reviewer 2 Report
I have read the manuscript entitled: “Enzymatic extraction and characterization of pectin from cocoa pod husks (Theobroma cacao L.) using Celluclast® 1.5L”. The manuscript is generally professionally written but is not ready to be published in its actual form. The main reason being the characterization of the pectin material reported. There is a critical feature in the characterization and that is Galacturonic acid and neutral sugars for it determines the structure of pectin and explains the resulting rheological and chemical properties. Chromatographic techniques are suggested to determine the galacturonic acid and neutral sugars profile. These new data would help understand how a low-methoxyl pectin is not reacting to calcium. Secondly, rheology is not fully assessed and neither storage nor loss moduli are presented.
The sole argument of a 2% higher yield is not strongly supporting the advantage of an enzymatic process to extract pectin in this study. The characterization with Chromatographic methods may well support a higher antioxidant, stronger gel, derived from the process proposed.
I strongly suggest the authors to complement their valuable data with Chromatography and small deformation oscillatory rheology. They may contact one of many authors out there that have already reported and ask for help. I really think that the subject is particularly relevant and deserves to be published once completed.
Author Response
We greatly appreciate the suggested considerations as well as the time to read our manuscript, it is of great importance to improve our academic and professional skills. We have read each one in detail and they have been broken down in Table 2 to provide an answer. Please see the attachment.

Reviewer 3 Report
The manuscript entitled "Enzymatic extraction and characterization of pectin from cocoa pod husks (Theobroma cacao L.) using Celluclast® 1.5L" optimized pectin rheological and physicochemical features typical of this biomaterial and provides an interesting alternative for the valorization of cocoa husks. The experiments appear to be well planned, results are interesting and correct, the ideas and methods are correct. In my mind, the manuscript is acceptable for publication in molecules. Line 82-84 Considering that there is a market tendency that demands products labeled as completely natural or biobased, there are combined efforts from the industry to achieve their production [24,25]. 25 Xizhe Fu, Tarun Belwal, Giancarlo Cravotto, Zisheng Luo. Sono-physical and sono-chemical effects of ultrasound: Primary applications in extraction and freezing operations and influence on food components. Ultrasonics-Sonochemistry, 2020 (60) 104726Author Response
Response to Reviewer 3 Comments
We greatly appreciate your suggestions, as they are of great importance to improve the manuscript and for the development of academic skills. The suggested reference was included in the manuscript. Changes can be verified in the new version of the manuscript.
Point 1: Line 82-84 Considering that there is a market tendency that demands products labeled as completely natural or biobased, there are combined efforts from the industry to achieve their production [24,25]. 25 Xizhe Fu, Tarun Belwal, Giancarlo Cravotto, Zisheng Luo. Sono-physical and sono-chemical effects of ultrasound: Primary applications in extraction and freezing operations and influence on food components. Ultrasonics-Sonochemistry, 2020
Response: The suggested reference was included in the manuscript. Changes can be verified in the new version of the manuscript.
Round 2
Reviewer 2 Report
No comments